# Experimental Research on PVDF Sensing Surface Characteristic Curve Applied to Topography Perception

**DOI:** 10.3390/mi11110976

**Published:** 2020-10-30

**Authors:** Zhen Yu, Jing-Xian Yu, Chen-Yang Zhang

**Affiliations:** 1Key Laboratory of Metallurgical Equipment and Control Technology of Ministry of Education, Wuhan University of Science and Technology, Wuhan 430081, China; yujingxian2021@163.com (J.-X.Y.); 15737345656@163.com (C.-Y.Z.); 2Hubei Key Laboratory of Mechanical Transmission and Manufacturing Engineering, Wuhan University of Science and Technology, Wuhan 430081, China

**Keywords:** PVDF array, topography self-perception, curve reconstruction, intelligent equipment, Ferguson curve

## Abstract

With the development of intelligent technology, it is of great significance to develop intelligent equipment with topography self-sensing function. The micro morphology perception technology applied to intelligent equipment is the key technology for development. In this paper, at first, topography perception theory based on the PVDF (Polyvinylidene Fluoride) technology is researched, then an experimental study is conducted to sense the characteristic points of the geometric curve of the preset topography surface used in the PVDF film, and then the Ferguson curve model is used to reconstruct the topography characteristic curve. The experimental results show that the reconstruction curve can truly reflect the features of the characteristic curve of the surface of the preset topography, and the feasibility of topography surface sensing technology by PVDF sensing technology is verified. The research provides technical support for the development of intelligent equipment with topography self-sensing function.

## 1. Introduction

In recent years, intelligent manufacturing and intelligent equipment have become research hotspots, and three-dimensional complex topography perception technology is one of the key technologies for the development of intelligent equipment [1,2,3,4,5]. Contact complex 3D (three dimensional) topography perception applied to intelligent structures is an important research direction in the field of intelligent manufacture; its key technology is to use appropriate sensors to sense and acquire the value of three-dimensional feature points of complex shapes, to perform data processing of the value of the sensed feature points. The characteristic curve of the characteristic surface of the three-dimensional shape are then reconstructed by using reverse engineering technology, followed by the reconstruction of the three-dimensional shape of the sensed shape. It has wide application prospects in aviation [6,7], medical [8], intelligent robot [9], and other fields.

At present, complex three-dimensional topography perception and reconstruction technologies applied to the research of intelligent structures include two types: non-contact shape perception technology and contact shape perception technology. Non-contact topography sensing technology mainly uses electromagnetic sensors [10], optical equipment [11], and other non-contact sensing equipment [12] to perform feature point sensing and data collection of the three-dimensional topography of the object, then, the data processing of the sensing feature points and the reconstruction of the three-dimensional topography surface characteristic curve and characteristic surface are carried out. The shortcomings of this non-contact sensing technology are: the interference from space-related fields is large, the sensing equipments are expensive, and the sensing cost is high. It is only used in some special fields. Contact three-dimensional topography sensing technology mainly uses flexible contact sensors such as optical fiber sensors [13,14,15], strain gauges [16,17], and others to sense the topography of objects [18]. The physical signal reflected by the sensor to reflect the deformation of the object is transformed into the curvature information of the three-dimensional topography characteristic curve [19,20], and then the curvature information is used to perform shape fitting to reconstruct the topography and realize the complex topography sensing. This technology needs to attach the sensor to the sensed three-dimensional body, and needs to deform the sensed object, so the key question is in choosing the sensing sensor.

Polyvinylidene Fluoride [21,22] (abbreviation: PVDF) is a piezoelectric polymer material that offers advantages of strong piezoelectric performance, wide frequency response, light weight properties, and good flexibility. It is widely used for excellent sensing characteristics and offers light flexibility characteristics to the PVDF piezoelectric film [23]. At present, the application research of PVDF mainly focuses on tactile perception [24], impact detection [25,26], bionic prosthesis [27], intelligent structure [28], motion monitoring [29], and others [30,31]. Because of its good flexibility and sensing characteristics for PVDF piezoelectric films, the material has obvious advantages with the use of PVDF for contact topography sensing application research.

In this paper, our goal is to use PVDF to develop smart equipment with tactile or pressure feedback. This research is an initial one on the application of PVDF in developing intelligent equipment, experimenting its output characteristics under pressure, and studying how to use PVDF array to improve its pressure perception accuracy and building theoretical models. On the premise of studying the principle of contact topography perception of PVDF piezoelectric film array, the mechanism of sensing the planar curve of PVDF is studied. An experimental study to sense the topography characteristic curve is then carried out, and the electrical signal sensed by discrete PVDF film arrays is used to collect and fit the topographic feature curve. The combination of theoretical research and experimental research provides a basis for the design of a self-sensing intelligent structure based on PVDF piezoelectric film array.

## 2. The Principle of Three-Dimensional Topography Characteristic Curve Sensing Based on PVDF Array

### 2.1. PVDF Piezoelectric Film Topography Sensing Mechanism

According to the piezoelectric effect of PVDF piezoelectric film, the piezoelectric equation of PVDF film is expressed as:(1)$$\{\begin{array}{c} \sigma_{i}=c_{ij}\varepsilon_{j}+e_{nj}E_{n}\\ D_{i}=d_{ij}\sigma_{j}+\varepsilon_{im}^{T}E_{m} \end{array}$$

In the equation, σ is stress value, E is electric field strength, D is surface charge density matrix, $\varepsilon^{T}$ is transposed matrix of dielectric constant matrix, c,d,e is piezoelectric strain constant matrix, i,m=1,2,3; j=1,2⋯6.

The piezoelectric equation reflects the functional relationship between the external electric field strength, the pressure on the film, and the charge generated on the film surface. The surface charge density matrix is superimposed by the external force and the influence of the applied electric field. When the electric field is zero and only the electrical boundary conditions are considered, the expression of the first type of piezoelectric equation of the PVDF piezoelectric film after ignoring the influence of the electric field is expressed as (2):(2)$$D_{i}=d_{ij}\sigma_{j}$$

In the equation, $D_{i}^{j}$ indicates the surface density of charge accumulation in the i direction when the force of j direction is applied (that is, the polarization in the i direction); $d_{ij}$ indicates piezoelectric constant generated in the i direction when the j direction is stressed, it indicates the polarization direction of the crystal, $\sigma_{j}$ indicates the stress per unit area when an external force is applied in the j direction.

When the pressure acts vertically on the PVDF surface, the charge concentration mainly lies in the polarization direction of the PVDF films, and the amount of charge generated by PVDF is calculated by Formula (3):(3)$$Q=d_{33}S\sigma_{33}=d_{33}F_{N}$$

In the equation, Q is charge generated in PVDF film; S is surface area of PVDF film; $F_{N}$ is pressure value acting on PVDF surface.

As can be seen from the above formula, the amount of charge perpendicular to the surface of the film is proportional to the positive pressure value.

### 2.2. Analysis of Three-Dimensional Shape Sensing Model Based on PVDF Piezoelectric Film

When studying the three-dimensional topography sensed by the PVDF arrays, the PVDF film needs to be placed on a soft and elastic substrate material to make a PVDF film sensor, then the PVDF film sensor is closely fitted on the curved surface and arranged in a linear array (shown in Figure 1). When the uniformly distributed load is acted on the PVDF film sensor array, charge is generated on the PVDF film sensor that is proportional to the amount of deformation with the deformation of the sensed topography, and the amount of charge generated is changed with the angle of the topography measured by the PVDF film sensors.

From Formula (3), for a single PVDF film, when F is acted on the surface perpendicularly, charge Q will be generated on the surface, and known from Figure 1, if the inclination angle of the inclined surface of the membrane unit is θ, and the local load of the external force is $\sigma_{0}$, when the PVDF film is close to the inclined surface, the load applied perpendicular to the PVDF film surface is expressed by Formula (4):(4)$$\sigma_{N}=\sigma_{0}\cos\theta$$

Let the PVDF piezoelectric film be square with side length a, due to the small size of the piezoelectric sheet, the force can be approximated as a uniform load, then the amount of charge generated on the PVDF surface is calculated by Formula (5):(5)$$Q=d_{33}\cdot F=d_{33}a^{2}\sigma_{N}=d_{33}a^{2}\sigma_{0}\cos\theta$$

Generally, any curve (non-straight line) has inflection points, for the PVDF film sensor arrays, the pressure at the curve inflection point is approximately the maximum value, so the force can be analyzed by calculating the tangent cosine of the contact point between the piezoelectric film and the curved surface. The denser the membrane per unit area, the more accurate the positive pressure obtained, and the more accurate the cosine value corresponding to the angle at this point. Taking the maximum pressure value approximately as the positive pressure value, there is:(6)$$\sigma =\sigma_{\max}=\underset{i=1\cdots n}{\max}\left[\sigma_{N}\right]$$

Then:(7)$$Q_{\max}=d_{33}\cdot F=d_{33}a^{2}\sigma_{N\max}=d_{33}a^{2}\cos\theta \approx d_{33}a^{2}\sigma_{0}$$

For other piezoelectric films with charge value less than the maximum value, its value is calculated by Formula (8):(8)$$Q=d_{33}a^{2}\cos\theta$$

Then the ratio can be obtained:(9)$$c=\frac{Q}{Q_{\max}}=\frac{d_{33}a^{2}\sigma \cos\theta }{d_{33}a^{2}\sigma }=\cos\theta$$

Then the angle value θ is calculated by Formula (10):(10)θ=arccosc

If the amount of charge is scalar and is always positive, then constant c is a constant positive value and the θ calculated is an acute angle, which can only reflect the inclination of the tangent curve at the patch on the surface, and cannot specifically reflect the concave and convex state of the curve. In order to reflect the positional relationship between two adjacent points, a standard needs to be taken to determine the relative size of the angle between each point; for this, the initial slope is assumed to be positive and the angle to be acute. When the maximum value is reached, it is considered that an extreme point has been reached; at this time, the slope should change sign, and the angle increases π/2 by the acute value. Therefore, every time a maximum value is reached, the slope changes accordingly. Assuming that the curve is strictly monotone in each monotone interval and the slope is positive, then the maximum value taken is the extreme value of the curve, that is, there is no case where the amount of charge is maximum but not extreme, the slope size described by this rule and the relationship between the adjacent slices conform to the actual morphological changes.

According to the established coordinate system, the approximate tilt angle of each type value point is obtained using the sensing characteristics of the PVDF piezoelectric film; then the relative position between the various types of value points is determined according to the geometric relationship. Figure 2 is a flow chart of the recursive algorithm for extracting value points based on the PVDF film sensor output signal.

Assuming that the size of the piezoelectric sheet is much smaller than the size of the object, when the PVDF sheet is attached to the surface of the object, the PVDF sheet is equally distributed on the object, as shown in Figure 3.

As seen in Figure 4, the curve tangent OC and AE are made, respectively, on the contact points O and A between the thin film microelement and the curve, and the two tangent lines are intersected at a point C, then, the ∠COB and ∠EAF are angles between the tangent and the horizontal line at the contact point between the film microelement and the curve; they are recorded as $\alpha_{1}$ and $\alpha_{2}$.

The rotation angle formed by the two tangent lines are thus $∠COF=∠COB-∠EAF=\alpha_{1}-\alpha_{2}$; it can be seen from the geometric relationship that the two tangent angles are equal to the central angle, that is: $∠ODA=\alpha_{1}-\alpha_{2}$. Let $\beta_{n}$ be the central angle wherein the PVDF film micro-element is in contact with the n curve contact point and its adjacent contact point, then:(11)$$\beta_{n}=\alpha_{n}-\alpha_{n+1}$$

As the curve between two points can be approximated as an arc, and the angle formed by the tangent and the two-point line is the chord angle, then $∠COA=\frac{\alpha_{1}-\alpha_{2}}{2}$; the angle formed by the string AO and the x axis is:(12)$$∠AOB=\gamma =\alpha_{1}-\frac{\alpha_{1}-\alpha_{2}}{2}=\frac{\alpha_{1}+\alpha_{2}}{2}$$

If $\gamma_{n}=\frac{\alpha_{n}+\alpha_{n+1}}{2}$, then the inclination of the two-point connection line can be approximately expressed, at the same time, According to the relationship between chord length and arc length, set the length of $A\overset{⌢}{B}$ to l; the chord length of $A\overset{⌢}{B}$ is d, the diameter of the circle O is ϕ, then:(13)$$d=\phi \sin\frac{l}{\phi }$$

In the present study, since the distance between the two piezoelectric film sheets is equal, the arc length between the midpoints of the two films is constant, and the corresponding radius is calculated by Equation (14):(14)$$r_{n}=\frac{l}{\beta_{n}}$$

Then:(15)$$d(\beta )=2\frac{l}{\beta }\sin\frac{\beta }{2}$$

The abscissa increment is:(16)$$\Delta x_{1}=d\cos\frac{\alpha_{1}+\alpha_{2}}{2}=d\cos\gamma_{1}$$

Then the coordinates of the first type point on the curve can be approximately expressed as $\left(d_{1}\cos\gamma_{1},d_{1}\sin\gamma_{1}\right)$, the same reason for the abscissa increment of each connection being $\Delta x_{n}=d_{n}\cos\gamma_{n}$; the ordinate increment is $\Delta y_{n}=d_{n}\sin\gamma_{n}$, then the abscissa of the first n piezoelectric sheet is expressed as:(17)$$x_{n}=d_{1}\cos\gamma_{1}+d_{2}\cos\gamma_{2}+d_{3}\cos\gamma_{3}+\cdots d_{n}\cos\gamma_{n}=\sum_{i=1}^{n}d_{n}\cos\gamma_{n}$$

The ordinate is expressed as:(18)$$y_{n}=d_{1}\sin\gamma_{1}+d_{2}\sin\gamma_{2}+d_{3}\sin\gamma_{3}+\cdots d_{n}\sin\gamma_{n}=\sum_{i=1}^{n}d_{n}\sin\gamma_{n}$$

Then the coordinate value of each approximate value point can be expressed as $\left(\sum_{i=1}^{n}d_{n}\cos\gamma_{n},\sum_{i=1}^{n}d_{n}\sin\gamma_{n}\right)$, as shown in Figure 4; l is the arc length of the two central points of the two film sheets. If the piezoelectric film is square and the side length is a and the interval size is b, then the distance between the two central points of the film is:(19)l=a+b

Substituting the PVDF arrays’ parameters into Equations (17) and (18), the coordinates of the value point measured by the PVDF film sensors can be expressed as:(20)$$\{\begin{array}{c} x=2(a+b)\sum_{i=1}^{n}\frac{1}{\beta_{n}}\sin\frac{\beta_{n}}{2}\cos\gamma_{n}\\ y=2(a+b)\sum_{i=1}^{n}\frac{1}{\beta_{n}}\sin\frac{\beta_{n}}{2}\sin\gamma_{n} \end{array}$$

The approximate relative position of the contact points between the PVDF film and the curve can be determined by Equation (20).

For the reconstruction of the topography curve, the vector function model is usually used to construct; as long as the relative position of the type value point does not change, the shape of the reconstructed surface topography does not change. A set of basis functions and associated coefficient vectors are used to build the mathematical models for most surfaces. Since the Ferguson curve is a special form of parameter cubic spline curve, and the parameter interval of each curve segment of Ferguson curve is taken as [0, 1], that is, the feature nodes are uniformly distributed, in this study, the Ferguson curve is used to reconstruct the characteristic curve of the sensed topography.

The Ferguson curve is expressed as Equation (21):(21)$$P(t)=a_{0}+a_{1}t+a_{2}t^{2}+a_{3}t^{3}$$

In the equation, t is parameter,$a_{0}$, $a_{1}$, $a_{2}$ and $a_{3}$ are vectors to be determined.

Let each type value node be $P_{i}$, then the curve segment $P_{i}P_{i+1}$$\left(0\leq t\leq t_{i+1}\right)$ can be expressed by Equation (22):(22)$$P_{i}(t)=P_{1}+P_{i}t+[\frac{3(P_{i+1}-P_{i})}{t_{i+1}^{2}}-\frac{2{P^{\prime }}_{i}}{t_{i+1}}-\frac{{P^{\prime }}_{i+1}}{t_{i+1}}]t^{2}+[\frac{2({P^{\prime }}_{i}-P_{i+1})}{t_{i+1}^{3}}+\frac{{P^{\prime }}_{i}}{t_{i+1}^{2}}+\frac{{P^{\prime }}_{i+1}}{t_{i+1}^{2}}]t^{3}$$

## 3. Experimental Research on Sensing of Three-Dimensional Topography Characteristic Curve Based on PVDF

In order to verify the feasibility of the reconstruction method of the three-dimensional topography characteristic curve based on a PVDF film sensor in this study, the research object using the fixed curve model is designed, the linear PVDF sensor array for sensing the curve model is designed, and the three-dimensional topography characteristic curve sensing experiment system is built. The three-dimensional topography characteristic curve sensing experiment hardware system is mainly composed of PVDF film sensors (Jinzhou Kexin Electronic Materials Co., Ltd., Jinzhou, China (thickness: 30 μm, length × width: 10 mm×10mm)), the sensed samples, and multi-channel charge amplifiers (Shenzhen Weijingyi Electronics Co., Ltd., Shenzhen, China, VK102-16), multi-function data acquisition card (US NI company, Austin, TX, USA, its model is USB-6210), computer, etc., the signal acquisition software is built with LabVIEW (education version) (US NI company, Austin, TX, USA).

In order to verify the reconstruction effect of the reconstruction curve, the selected curve needs to have a certain symmetry. In this study, the sinusoidal curve is selected as the preset curve for the experiment, and the characteristic curve of the measured topography is set to:(23)$$y=40\sin(\frac{\pi }{100}x)$$

The width of the sample curve of the measured three-dimensional topography is 5mm, the highest point $y_{\max}=40\mathrm{mm}$, and the circumference of the sensed curve is 13.3mm.

Because polyethylene terephthalate (PET) has excellent high and low temperature resistance, electrical insulation, adhesion, radiation resistance, medium resistance, and other characteristics, in this study, a PET film with a thickness of 125 μm is used as the base material, and the PVDF film is arranged at equal intervals; the sensing samples are curved convex modules and curved concave modules made of acrylic sheet (as shown in Figure 5 for the experimental system). Vertical force is then evenly applied above the module to make the PVDF film fully stressed.

The experimental process is as follows: lay a linear PVDF film sensor array on the curved sample model and apply vertical force evenly. The charge output is generated on the PVDF sensor array due to the PVDF sensor array deform with the module being sensed. The data acquisition card is used to collect and output the output voltage signal to the computer, and through the steps of signal processing, algorithm calculation, and graphic display, the topography curve of the sensed sample module is finally displayed. The voltage signal waveform of PVDF collected by LabVIEW is shown in Figure 6.

The measured curve of the experiment shows the action process and the withdrawal process of the uniform pressure during the experiment. The input signal applied is the pressure value, the voltages measured show peaks and the trenches at different points for the applied signal. This is a waveform of the loading and unloading process, and there is vibration interference during the loading and unloading process. The tip of the curve indicates the voltage value at the point of maximum contact force, and the decreasing curve indicates the process of contact force cancellation. Due to the morphological error of the sensed sample model, the contact force of each PVDF film is different, so the force cancellation process and recovery time of each test point are also different. The peak values of the different curves in Figure 6 represent the voltage values corresponding to the contact forces at different positions; the voltage data at the peak is derived in Table 1. Here, the peak voltage in Figure 6 should theoretically correspond to the one in Table 1, but it can be seen from the figure that there are errors, such as 3 and 5. The peak points in Figure 6 are quite different, while in Table 1, the peak values of the two are very close; the main reason for Table 1 is the data derived from the software system after the experiment, while Figure 6 is obtained by screenshots of the display during the experiment. There is an experimental process error between them.

Importing the experimental data into MATLAB Demo software (version 2018, American MathWorks company), the curve is reconstructed using the given curve reconstruction algorithm; the curve reconstruction result is shown in Figure 7.

Through the experimental analysis of the reconstruction curve, the *X*-axis error is relatively large, mainly because the size of the experimental sample is not accurate, the number of sensors in the PVDF array is small, and there is a lateral cumulative error. Moreover, this experiment focuses on the experimental results in the Y direction, while the size is in the X direction. It is measured by a scale, and when the piece arranged is not completely arranged on the theoretical value point, it can be seen that the reconstructed curve is basically close to the theoretical curve, which verifies the feasibility of the PVDF morphology detection method. However, there is still some reconstruction error—the error accumulation of the curve in the x axis direction is more obvious, the coordinates of the end of the curve are (79.76 mm, 4.14 mm), the error reaches the maximum, and the theoretical analysis shows that the error is 20.66 mm. The main reason for the error is that the errors generated during the experimental process are constituted by the curve module manufacturing error, data measurement error, data processing error, and curve model reconstruction principle error, and so on. In theory, as the PVDF film size decreases, the curve reconstruction accuracy will be further improved. How to improve the accuracy of three-dimensional topography sensing based on PVDF and reduce the error of topography sensing and topography reconstruction is the focus of the next research.

## 4. Error Analysis of Shape Sensing Experiment Using PVDF Array

In order to quantitatively analyze the shape reconstruction error, use the root mean square error (*RMSE*) to describe:(24)$$RMSE=\sqrt{\frac{\sum_{i=1}^{n}\left({\left(x_{i}^{\prime }-x_{i}\right)}^{2}+{\left(y_{i}^{\prime }-y_{i}\right)}^{2}\right)}{n}}$$
where, $\left(x_{i}^{\prime ;},y_{i}^{\prime }\right)$ is the value point at a point on the reconstructed surface, (x,y) is actual measuring point, n indicates the total number of sampling points. $x_{i}$, $y_{i}$ are original data, $x_{i}^{\prime }$, $y_{i}^{\prime }$ are reconstructed curve data. Set different numbers of sampling points on the curve in turn, to analyze the reconstruction error, and obtain the data table of reconstruction error and the number of sampling points (Table 2).

Figure 8 is the error curve.

The fitted curve is:(25)$$f(x)=4.947\times e^{0.4013x}+0.04717$$

It can be seen from Figure 8, theoretically, that the error of the fitting curve of this example is approximately exponentially related to the number of model value points. Overall, the reconstruction error decreases as the number of sampling points increases, especially if the sampling points are less than 14. This is because the more the sampling points, the more measurement information is obtained, and less measurement information lost between two discrete points; thus, the measurement results are more accurate. However, when the number of PVDF increases to 14, the reconstruction error decreases very little; at this time, there is more redundant measurement information, and the influence of the number of sensors on the reconstruction error is close to the limit; it is, thus, more difficult to obtain better measurement results.

In order to show a regular pattern of error more clearly, the error curve with the number of model value points is drawn (seen Figure 9 and Figure 10).

When the number of model value points is 19, the error of abscissa and ordinate varies with the node, as seen in Figure 11.

As can be seen from the figure, the abscissa error is much larger than the abscissa error. The abscissa error is approximately proportional to the increase in nodes, and the error accumulation is more obvious. The ordinate error has a certain volatility. It can be seen that there is a certain relationship with the curvature of the node location—when it is near the maximum point, the error reaches the maximum.

## 5. Conclusions

In this paper, an experimental study of the characteristic curve sensing of the sensed sample is carried out based on the theory of PVDF curve topography sensing. During the experiment, a sample module was designed and fabricated, a PVDF film sensor was fabricated, charge amplifier and data acquisition card were selected, an experimental scheme designed, and the experimental process carried out. In the experiment, LabVIEW software was used to collect experimental data, which was imported into MATLAB to reconstruct a three-dimensional topography characteristic curve of the sensed sample. The experimental research found that the use of PVDF can be used for micro-topography sensing, and can obtain micro-topography characteristics; however, sensing accuracy is related to the placement point and placement density of the sensor array, as well as the size of the PVDF sensor. The experimental results show that the experiment reconstruction curve reflects approximately the shape of the preset curve, which verifies the feasibility of the PVDF contact topography sensing method, and provides a research basis for the application of PVDF in the design of topography self-sensing intelligent structures. In future steps, the thickness of the PVDF film (30 um) will be influential; we will have to consider the thickness of PVDF, establish a more accurate mathematical model, conduct experimental research, carry out experimental corrections and change the experimental conditions for the PVDF sensor array micro-topography sensing, and carry out error analysis affecting the micro-topography to obtain a more accurate micro-topography experiment plan.

## Figures and Tables

**Figure 1 micromachines-11-00976-f001:**
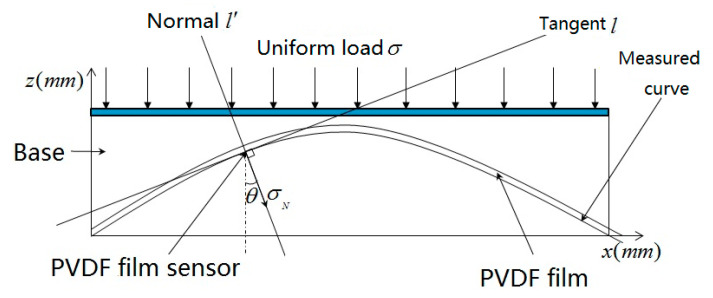
Schematic diagram of PVDF topography sensing structure.

**Figure 2 micromachines-11-00976-f002:**
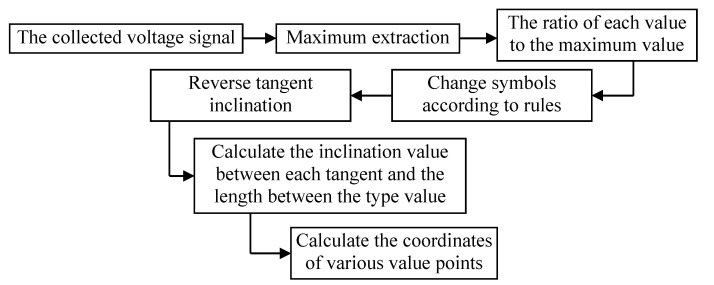
Flow chart of the recursive algorithm for extracting value points based on PVDF film sensor output signal.

**Figure 3 micromachines-11-00976-f003:**
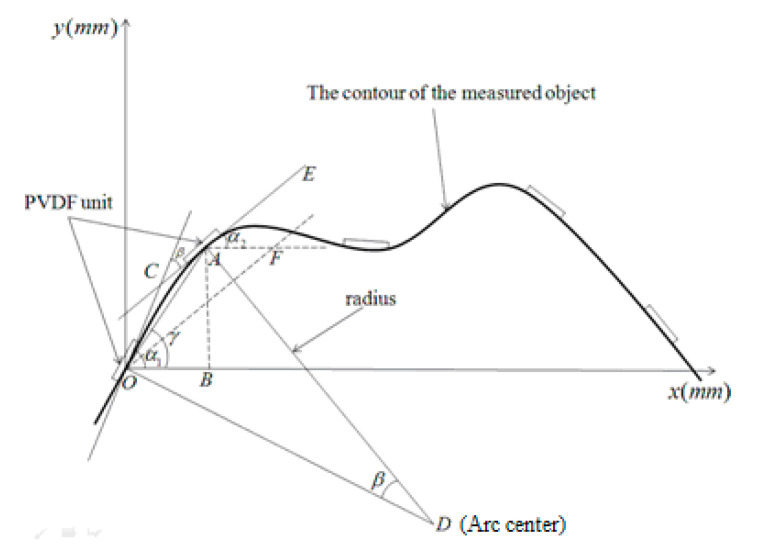
Geometric analysis of curve fitting.

**Figure 4 micromachines-11-00976-f004:**
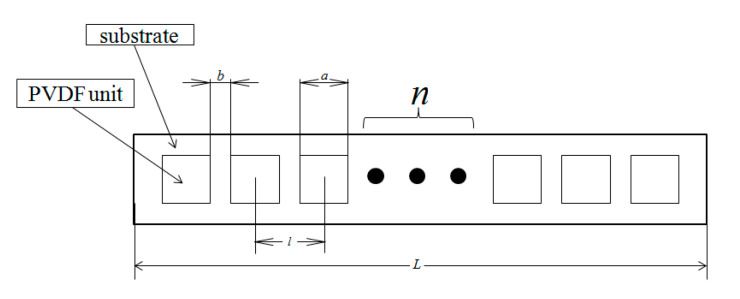
Layout of the PVDF unit linear array for topographical sensing.

**Figure 5 micromachines-11-00976-f005:**
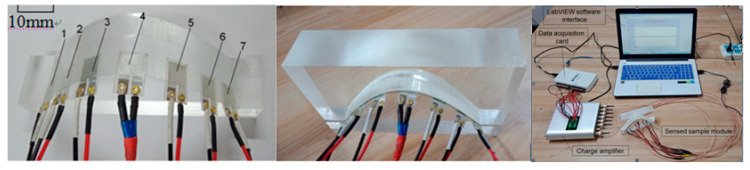
Topography sensing experiment system.

**Figure 6 micromachines-11-00976-f006:**
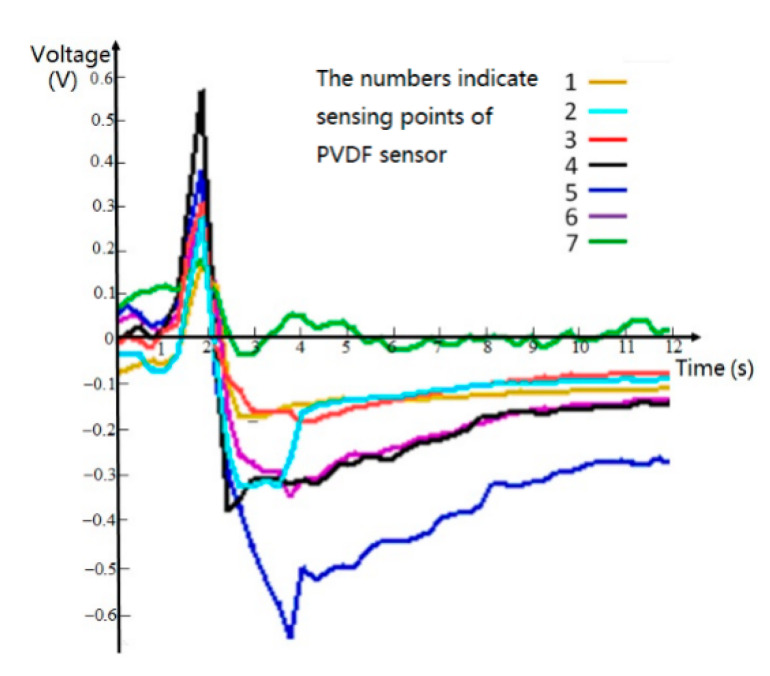
Output voltage waveform diagram of PVDF under pressure using LabVIEW.

**Figure 7 micromachines-11-00976-f007:**
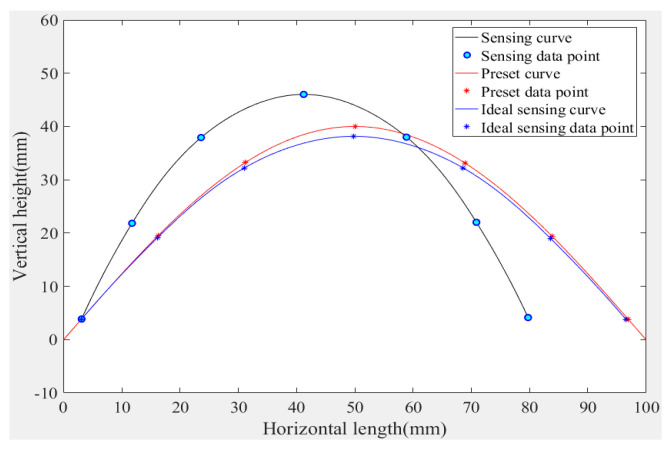
Reconstructed characteristic curve of morphology of the sensed module by experiment.

**Figure 8 micromachines-11-00976-f008:**
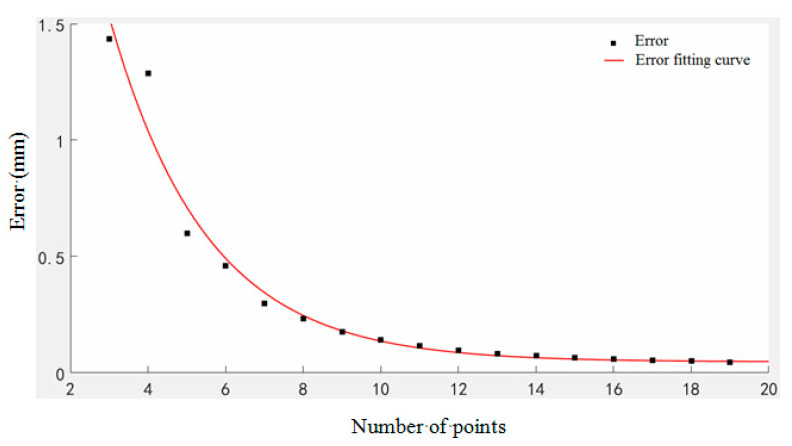
The relationship between the number of model value points and reconstruction error.

**Figure 9 micromachines-11-00976-f009:**
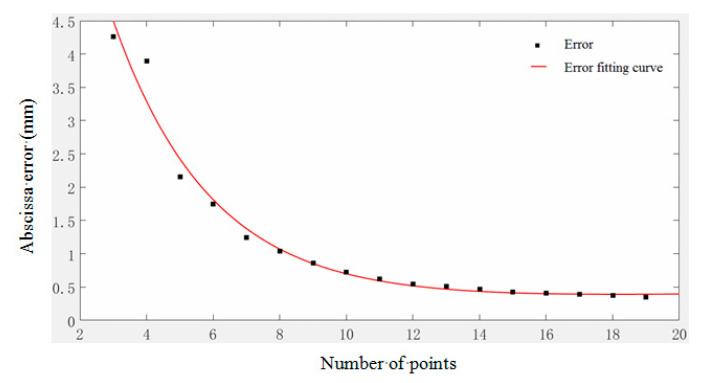
Variation curve of abscissa error with the number of model value points.

**Figure 10 micromachines-11-00976-f010:**
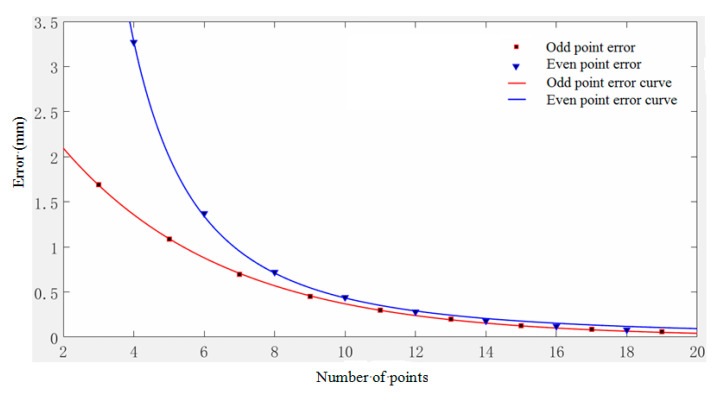
Variation curve of ordinate error with the number of model value points.

**Figure 11 micromachines-11-00976-f011:**
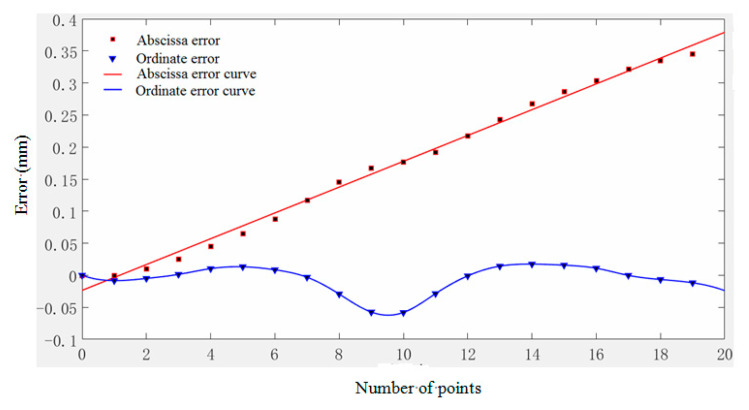
The law of error changes with the change of node coordinates.

**Table 1 micromachines-11-00976-t001:** Voltage value of each test point of the sensed sample module.

PVDF Number	1	2	3	4	5	6	7
Voltage(V)	0.1918	0.3076	0.3767	0.5784	0.3805	0.3123	0.1993

**Table 2 micromachines-11-00976-t002:** Table of error values caused by different numbers of value points.

Number of Points	Root Mean Square Error of Curve Fitting (mm)	X Coordinate Error (mm)	Y Coordinate Error (mm)
3	1.4364	4.26	1.69
4	1.2871	3.90	3.27
5	0.5999	2.16	1.09
6	0.4608	1.75	1.37
7	0.2991	1.25	0.70
8	0.2335	1.04	0.72
9	0.1781	0.86	0.45
10	0.1431	0.73	0.44
11	0.1160	0.62	0.30
12	0.0966	0.55	0.28
13	0.0841	0.51	0.20
14	0.0742	0.47	0.18
15	0.0661	0.43	0.13
16	0.0603	0.41	0.12
17	0.0548	0.39	0.09
18	0.0515	0.38	0.08
19	0.0459	0.35	0.06

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
