# Peer review of "Experimental Research on PVDF Sensing Surface Characteristic Curve Applied to Topography Perception"

_micromachines, 2020, doi:10.3390/mi11110976_

Round 1

Reviewer 1 Report

Very preliminary results, according to the error in figure 9. I would suggest to improve the results to consider publication.

Some additional comments that I hope help.

  1. Introduction.

How novelty is your approach? Is this the first time PVDF is used with this purpose?

  1. The principle…

Is this the common approach for this technique?

Is it based on previous works by other authors?

  1. Experimental…

Why do you require PET?. You could only use PVDF, with metal on both sides. And with a scriber defined as many sensors as you need.

The cables to contact PVDF should be changed, to improve the flexibility of the set-up.

Which is the minimum feature size you approach can sense?

What about more complicated samples to sense?

Author Response

Comments and suggestions for authors

Very preliminary results, according to the error in figure 9. I would suggest to improve the results to consider publication.Some additional comments that I hope help.

1.Introduction.

How novelty is your approach? Is this the first time PVDF is used with this purpose?

Reply: Our goal is to use PVDF to develop smart equipment with tactile or pressure feedback. In this research, few people are currently doing it. This research is the initial research on the application of PVDF to develop intelligent equipment, experimenting its output characteristics under pressure, and how to use PVDF array to improve its pressure perception accuracy, and building theoretical models. There are not many scholars doing this research.

2.The principle…

Is this the common approach for this technique? Is it based on previous works by other authors?

Reply:The use of PVDF arrays to sense pressure is indeed a common research method, but there are almost no research results in this area, such as the geometric parameters of the PVDF array, the correlation model between the layout method and the experimental results, and the accuracy of its sensing pressure.

  1. Experimental…

Why do you require PET? You could only use PVDF, with metal on both sides. And with a scriber defined as many sensors as you need.

Reply: PET (polyethylene terephthalate) is a commonly used flexible insulating material, and most electronic devices use this material. And PET has excellent high and low temperature resistance, electrical insulation, adhesion, radiation resistance, medium resistance and other characteristics This feature is also used here. Using PET as the substrate is to reduce the error of PVDF sensing pressure. In the PVDF arrangement, there is a certain error in the use of the scriber. After the feasibility is verified, the PVDF arrangement will be more accurate.

  1. The cables to contact PVDF should be changed, to improve the flexibility of the set-up.

Reply: Yes, thanks, the PVDF cable restricts its flexibility and will be further improved in the future.

  1. Which is the minimum feature size you approach can sense? What about more complicated samples to sense?

Reply: Sorry, this research needs further in-depth research, and its sensing resolution is also the content of further research. The purpose of this research is to construct and verify the theoretical model of PVDF array geometric parameters, layout method and output.

For the problems of complex shape perception, first of all, the PVDF array used for topography sensing is only to sense the topography with little unevenness. For complex topography features, the research method adopted is: to reduce the size of a single PVDF sensor, to reduce the spacing between sensors, and to make as many sensor arrays as possible to cover the measured topography.

Reviewer 2 Report

The English style is repetitive and sometimes difficult to follow, but the ideas are adequately communicated.

The process going from voltage signal to coordinates as outlined in Figure 2 is not clear.  Uncertainties in the final coordinates are not addressed.

The test setup may match the theoretical description well, with sensor squeezed between two complementary shaped blocks to provide a uniform load.  However it seems like an unlikely practical situation, is there an application where this might be the case?

Results appear to be from a single measurement?  This measurement likely depends on accurate matching of sensors as well and loading and unloading techniques.  Practical issues such as these are not discussed.

From Figure 8 it looks like the peak voltages do not agree with Table 1, in particular peak values of sensor 3 and 5.  Are these different measurements or is Figure 8 unclear?

The key results in fig 9 are not well explained in the caption or the subsequent text.  In particular the large mismatch in x coordinates is very noticeable and should be addressed in some detail.

Conclusion discusses ‘micro-topography’ which is not demonstrated in the paper.  A significant portion of the conclusion is speculation on future research activities.

Author Response

Comments and suggestions for authors

  1. The English style is repetitive and sometimes difficult to follow, but the ideas are adequately communicated. The process going from voltage signal to coordinates as outlined in Figure 2 is not clear. Uncertainties in the final coordinates are not addressed.

Reply: Thank you for your suggestion, some language expression problems have been revised. Figure 2 is a flowchart of this research process. Explaining the steps of this research, the specific research methods and detailed process, as well as the problems to be solved, and how to verify the theoretical model based on the experimental results, mainly in the following analysis.

  1. The test setup may match the theoretical description well, with sensor squeezed between two complementary shaped blocks to provide a uniform load. However it seems like an unlikely practical situation, is there an application where this might be the case?Results appear to be from a single measurement? This measurement likely depends on accurate matching of sensors as well and loading and unloading techniques. Practical issues such as these are not discussed.

Reply: Since experiments are used to verify the rationality of the mathematical model of curve fitting constructed above, the ideal design is adopted as much as possible in the experimental design, including the curve modules used in the experiment, which adopt the standard shape curve Shape. This is to minimize the influence of errors in the experiment process on the verification of the mathematical model. Once the verified model is reasonable, then this mathematical model can be used to fitted to perceive the topographic features according to the output results sampled the PVDF array for any irregular topography.  Through this experiment, it is verified that the curve fitting mathematical model constructed above is feasible, which is also the purpose and significance of this experiment.

  1. From Figure 8 it looks like the peak voltages do not agree with Table 1, in particular peak values of sensor 3 and 5. Are these different measurements or is Figure 8 unclear?

Reply: Thank you , the peak voltage in Figure 8 should theoretically correspond to the one in Table 1, but it can be seen from the figure that there are errors, such as 3 and 5, the peak points in Figure 8 are quite different, while in Table 1, the peak values of the two are very close, the main reason is that Table 1 is the data derived from the software system after the experiment, while Figure 8 is obtained by screenshots of the display during the experiment, there is an experimental process error between them.

  1. The key results in fig 9 are not well explained in the caption or the subsequent text. In particular the large mismatch in x coordinates is very noticeable and should be addressed in some detail.

Reply: This research is mainly to study the principle level, the X-axis error is relatively large, mainly because the size of the experimental sample is not accurate, and the number of sensors in the PVDF array is small, and there is a lateral cumulative error. Moreover, this experiment focuses on the experimental results in the Y direction, while the size in the X direction, it is measured by a scale, and when the piece arranged is not completely arranged on the theoretical value point, for large errors in the x-axis direction, a correlation analysis of the error is required to obtain a method to reduce the error. This part requires follow-up research, and may involve intelligent algorithms, etc., which is difficult to complete all at once. To solve the problem, in-depth research will be conducted later.

  1. Conclusion discusses ‘micro-topography’ which is not demonstrated in the paper. A significant portion of the conclusion is speculation on future research activities.

Reply: Yes, Micro-topography is the purpose of this research. The purpose of using PVDF array for topography sensing is to use PVDF's fine sensing mechanism to perceive micro topography, this is also the direction that needs to be studied in the next step.

Reviewer 3 Report

The authors have defined a method for topography sensing and shown its working using experimental data. However, the paper is rather light in terms of highlighting the importance of this method to current results, as well as other existing methods. Some comments are as follows: Please define PVDF in full form where it is used first (in abstract), as it is defined in the introduction. Lines 69-78: There is no need to exclusively describe piezoelectric effect in general. This is a common transduction mechanism. Also, consider rephrasing the term “positive” PE. This word is redundant. Equation 1: This is a general piezoelectric equation, not specifically for PVDF. Change the text above it. Sections 2 can be merged into section 3 as a suggestion since section 2 only describes general piezoelectrics. Figures 5, 6 and 7 should be merged into a single figure. Figures 8 and 9: Quality of the figures can be improved, please use bigger font for legend and axes. Figure 8: Why do the voltages show peaks and trenches at different points for applied signal? Is this behavior expected from the testing? What is the input signal that is applied? Please explain with more clarity. Please address the following general qs: Could a section explaining the fabrication (or development) of the PVDF array be included (for eg, how is the film coated on the array, patterning, etc)? How are the electrodes connected to the array elements? How do the results presented in the paper compare to similar sensing technologies for both contact and non-contact sensing? Perhaps an inclusion of a table of comparison might add more relevance to the experiments. As an add-on to the earlier question, what is the real motivation behind this sensing platform and how does it improve on existing technology? Please explain with regards to the results. What is the error percentage in the reconstructed curves? Please quantify the results better. How does the measured data compare with the calculated values as the equation 22? As a general comment, the authors have merely just stated the results. The main purpose of a paper should be highlighting the importance of the results and the motivation behind the work. This aspect is missing from the manuscript, which should be included or brought to the readers’ attention in a better way. Font size is not uniform in the conclusion, please fix it.

Author Response

Comments and suggestions for authors

  1. The authors have defined a method for topography sensing and shown its working using experimental data. However, the paper is rather light in terms of highlighting the importance of this method to current results, as well as other existing methods. Some comments are as follows: Please define PVDF in full form where it is used first (in abstract), as it is defined in the introduction.

Reply: Thanks, I have revised it.

  1. Lines 69-78: There is no need to exclusively describe piezoelectric effect in general. This is a common transduction mechanism. Also, consider rephrasing the term “positive” PE. This word is redundant. Equation 1: This is a general piezoelectric equation, not specifically for PVDF. Change the text above it.

Reply: Thanks, Your opinion is very reasonable, I have deleted and revised this part of the content.

  1. Sections 2 can be merged into section 3 as a suggestion since section 2 only describes general piezoelectrics.

Reply: Thanks ,I have revised it.

  1. Figures 5, 6 and 7 should be merged into a single figure.

Reply: Thanks, the figure has been merged.

  1. Figures 8 and 9: Quality of the figures can be improved, please use bigger font for legend and axes. Figure 8: Why do the voltages show peaks and trenches at different points for applied signal? Is this behavior expected from the testing? What is the input signal that is applied? Please explain with more clarity.

Reply: Figure 6: the voltages measured show peaks and trenches at different points for applied signal, this is a waveform of the loading and unloading process, and there is vibration interference during the loading and unloading process. The input signal applied is the pressure value.

  1. Please address the following general qs: Could a section explaining the fabrication (or development) of the PVDF array be included (for eg, how is the film coated on the array, patterning, etc)?How are the electrodes connected to the array elements? How do the results presented in the paper compare to similar sensing technologies for both contact and non-contact sensing? Perhaps an inclusion of a table of comparison might add more relevance to the experiments.

Reply: Thanks, the use of PVDF arrays to make specific sensors that can be applied in practice is the next step. Each sensor unit of the PVDF array has electrode connections, all the electrodes of the PVDF array are connected to the NI multi-channel input unit. The measurement of microscopic morphology for the PVDF array sensor belongs to contact measurement, the sensing principle is different from other non-contact measurement of optical sensors, there is no analysis of the results of related researches, there is no comparison analysis table for the time being.

  1. As an add-on to the earlier question, what is the real motivation behind this sensing platform and how does it improve on existing technology?

Reply: The purpose of this research is to build a mathematical model of the PVDF array sensing topography through fitting using the model value points used by the PVDF array, and to verify the rationality of the model through experiments. The next step is to design and make the PVDF array sensing system for  micro-topography sensing, the purpose of the research is to develop smart equipment with micro-topography perception.

  1. Please explain with regards to the results. What is the error percentage in the reconstructed curves? Please quantify the results better. How does the measured data compare with the calculated values as the equation 22? As a general comment, the authors have merely just stated the results.

Reply: I have added the error analysis content of the shape perception experiment to explain.

  1. Error analysis of shape sensing experiment using PVDF array

In order to quantitatively analyze the shape reconstruction error, use the root mean square error (RMSE) to describe:

           (24)

Where, is the value point at a point on the reconstructed surface,  is actual measuring point, indicates the total number of sampling points. 、are original datu,、 are reconstructed curve datu. Set different numbers of sampling points on the curve in turn, to analyze the reconstruction error, and obtain the data table of reconstruction error and the number of sampling points in Table 2.

Table 2 Table of error values caused by different number of value points

Number of points

Root mean square error of curve fitting(mm)

X coordinate error(mm)

Y coordinate error(mm)

3

1.4364

4.26

1.69

4

1.2871

3.90

3.27

5

0.5999

2.16

1.09

6

0.4608

1.75

1.37

7

0.2991

1.25

0.70

8

0.2335

1.04

0.72

9

0.1781

0.86

0.45

10

0.1431

0.73

0.44

11

0.1160

0.62

0.30

12

0.0966

0.55

0.28

13

0.0841

0.51

0.20

14

0.0742

0.47

0.18

15

0.0661

0.43

0.13

16

0.0603

0.41

0.12

17

0.0548

0.39

0.09

18

0.0515

0.38

0.08

19

0.0459

0.35

0.06

Figure 8 is the error curve.

Figure 8. The relationship between the number of model value points and reconstruction error

The fitted curve is:

          (25)

It can be seen from the Figure 8, theoretically, the error of the fitting curve of this example is approximately exponentially related to the number of model value points. Overall, the reconstruction error decreases as the number of sampling points increases, especially if the sampling points are less than 14, this is because the more sampling points, the more measurement information obtained, and the less measurement information lost between two discrete points, so that the measurement results are more accurate. However, when the number of PVDF increases to 14, the reconstruction error decreases very little, at this time, there is more redundant measurement information, and the influence of the number of sensors on the reconstruction error is close to the limit, and it is more difficult to obtain better measurement results.

In order to show the regular pattern of error more clearly, the error curve with the number of model value points is drawn(seen Figure 9 and Figure 10).

Figure 9.  Variation curve of abscissa error with the number of model value points

Figure 10.  Variation curve of ordinate error with the number of model value points

When the number of model value points is 19, the error of abscissa and ordinate varies with the the node is seen in Figure 11.

Figure 11. The law of error changes with the change of node coordinates

As can be seen from the figure, the abscissa error is much larger than the abscissa error. The abscissa error is approximately proportional to the increase of nodes, and the error accumulation is more obvious. The ordinate error has a certain volatility, it can be seen that there is a certain relationship with the curvature of the node location, when it is near the maximum point, the error reaches the maximum.

  1. The main purpose of a paper should be highlighting the importance of the results and the motivation behind the work. This aspect is missing from the manuscript, which should be included or brought to the readers’ attention in a better way. Font size is not uniform in the conclusion, please fix it.

Reply: The paper has been revised, and the purpose of the research and the content of the follow-up need to be studied are emphasized in the corresponding revised position.

Round 2

Reviewer 2 Report

This paper presents only preliminary experimental results, however the authors appear to have addressed the main reviewer comments.

Reviewer 3 Report

Authors have satisfactorily addressed the issues in the revision. Thanks!

This manuscript is a resubmission of an earlier submission. The following is a list of the peer review reports and author responses from that submission.

Round 1

Reviewer 1 Report

This manuscript describes the theoretical and experimental design of a piezoelectric sensor system for topographical measurements. There is some novelty to this design, however the manuscript needs to be improved in several regards before it is acceptable for publication.

  1. The authors will need to clearly motivate the scientific interest of this sort of measurement system. The first impression is that a system which requires a force to be applied by a 'negative' copy of the topography to be measured would be of very limited practical use.
  2. P. 2 row 74: what is "the internal electrode axis"?
  3. P. 4 row 145-147: "when the maximum value is reached, it is considered that an extreme point has been reached, at this time the slope should change sign, and the angle increases pi / 2
    147 by the acute value". Again, this assumption severely limits the practical use of the system, as it can not be used to measure topographies where a plateau is followed by a second or more maxima.
  4. P. 5 figure 3: What is the "Center of mind"?
  5. P. 7 row 241: "polyimide film (PET)". Polyimide is typically abbreviated PI, and poly(ethylene terephtalate) is typically abbreviated PET. Which material was used in the present work?
  6. P. 8 Figure 8, results. How reproducible is the output voltage? How many experiments were performed? How predictable is the max output voltage?
  7. The authors are encouraged to perform reference measurements with the linear sensor array on a horizontal surface. Is the response identical from all sensors?
  8. The conclusions is a summary of the project, rather than an interesting summary of conclusions.
  9. The English language needs to be greatly improved.

Reviewer 2 Report

Authors have presented PVDF sensing surface characteristic curve applied to topography perception. My comments and concerns are as follows:

  1. The manuscript lacks novelty and originality to justify a research publication.
  2. The reported study has been well researched by a number of researchers in the past. 
  3. Very poorly written article in terms of English language and technical matter.
  4. Various typos and repetitive words/sentences.
  5. Theory is mostly book knowledge and does not contribute to the article. This is well known.

The experimental part can be considered as a small project to study, but does not by any mean justify a research article.

Reviewer 3 Report

The authors present a PVDF array made of several PVDF films serving as surface characteristic curve sensor, and both theoratical and experimental investigations are carried out. The research topic of topography perception based on piezoelectricity is interesting, but the scientific merit of this work is not clear. Moreover, several major challenges must be addressed before the consideration of publication. 1. Topography perception, especially topography characteristic curve reconstruction, is significant. The authors proposed a PVDF array to realize this purpose. However, this PVDF array is made of several SEPARATED PVDF films instead of a patterned PVDF film. So, one of critical challenge for this method is how to define and control the gap, the number, the relative position of PVDF films? And, all of these parameters are essential to affect the resolution ratio and precision of topograph perception. Why the authors did not pattern a single PVDF film with several pre-designed arrays (it is easy to be realized by using laser cutting or other fabrication process, actually). 2. It is good to have the theoratical analysis for modeling the developed device. However, one critical question is the absence of the thickness of PVDF films. Without the consideration of PVDF film thickness, the theoratical model and equations are not reasonable. Compared with the 125 um PET substrate, and the thickness of PVDF film (30 um) can not be ignored. 3. The format of references should be unified. The scale bars should be added to Fig. 5 and 6.